# SSL: A Self-similarity Loss for Improving Generative Image Super-resolution

### Du Chen*
Department of Computing,
The Hong Kong Polytechnic University,
Hong Kong, China
OPPO Research Institute,
Shenzhen, China
csdud.chen@connect.polyu.hk

### Zhengqiang Zhang*
Department of Computing,
The Hong Kong Polytechnic University,
Hong Kong, China
OPPO Research Institute,
Shenzhen, China
zhengqiang.zhang@connect.polyu.hk

### Jie Liang
OPPO Research Institute,
Shenzhen, China
liangjie3@oppo.com

### Lei Zhang†
Department of Computing,
The Hong Kong Polytechnic University,
Hong Kong, China
OPPO Research Institute,
Shenzhen, China
cslzhang@comp.polyu.edu.hk

## Abstract

Generative adversarial networks (GAN) and generative diffusion models (DM) have been widely used in real-world image super-resolution (Real-ISR) to enhance the image perceptual quality. However, these generative models are prone to generating visual artifacts and false image structures, resulting in unnatural Real-ISR results. Based on the fact that natural images exhibit high self-similarities, i.e., a local patch can have many similar patches to it in the whole image, in this work we propose a simple yet effective self-similarity loss (SSL) to improve the performance of generative Real-ISR models, enhancing the hallucination of structural and textural details while reducing the unpleasant visual artifacts. Specifically, we compute a self-similarity graph (SSG) of the ground-truth image, and enforce the SSG of Real-ISR output to be close to it. To reduce the training cost and focus on edge areas, we generate an edge mask from the ground-truth image, and compute the SSG only on the masked pixels. The proposed SSL serves as a general plug-and-play penalty, which could be easily applied to the off-the-shelf Real-ISR models. Our experiments demonstrate that, by coupling with SSL, the performance of many state-of-the-art Real-ISR models, including those GAN and DM based ones, can be largely improved, reproducing more perceptually realistic

image details and eliminating many false reconstructions and visual artifacts. Codes and supplementary material are available at https://github.com/ChrisDud0257/SSL

## CCS Concepts

• **Computing methodologies → Reconstruction**.

## Keywords

image super-resolution, generative adversarial networks, generative diffusion models, self-similarity loss

**ACM Reference Format:**
Du Chen, Zhengqiang Zhang, Jie Liang, and Lei Zhang. 2024. SSL: A Self-similarity Loss for Improving Generative Image Super-resolution. In *Proceedings of the 32nd ACM International Conference on Multimedia (MM '24), October 28–November 1, 2024, Melbourne, VIC, Australia.* ACM, New York, NY, USA, 10 pages. https://doi.org/10.1145/3664647.3680874

## 1 Introduction

Image super-resolution (ISR) is a fundamental problem in low-level vision. Given a low-resolution (LR) input, ISR aims to recover its high-resolution (HR) counterpart with high fidelity in contents, which has a wide range of applications in digital photography [28], high definition display [83], medical image analysis [27], remote sensing [34], *etc.* Starting from SRCNN [13], various convolutional neural network (CNN) based methods have been proposed to improve the ISR performance, such as residual connections [22, 31, 39, 42], dense connections [91] and channel-attention [23, 90]. Recently, some transformer-based ISR methods [7–9, 17, 40, 89] have also emerged and demonstrated more powerful performance.

In the early stage, researchers usually employed simple degradations, such as bicubic downsampling and downsampling after Gaussian smoothing, to synthesize the LR-HR training pairs, while focusing on the study of ISR network design. However, the image degradations in real-world are much more complex, and the ISR

*Both authors contributed equally to the paper
†Corresponding author

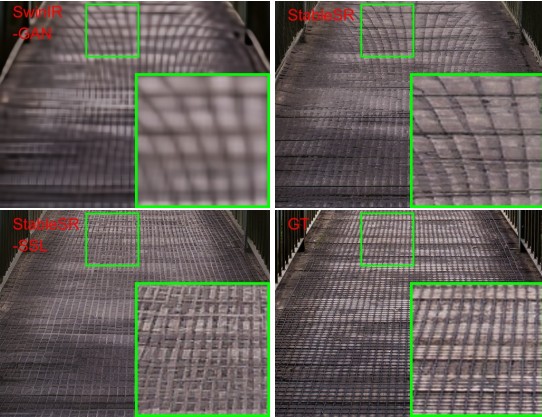

**Figure 1: From left to right and top to bottom: the Real-ISR results generated by SwinIRGAN [40], StableSR [63], our SSL guided StableSR and the ground-truth (GT) image. SwinIR-GAN produces over-smoothed and wrong results, while StableSR produces more details but with false structures and artifacts. Our SSL guided StableSR generates more faithful details while suppressing much the artifacts.**

models trained on those simple synthetic data can hardly be generalized to real-world applications. Therefore, in recent years many works have been done on real-world ISR (Real-ISR), aiming to obtain perceptually realistic ISR results on degraded images in real-world scenarios [10, 19, 36–38, 45, 53, 64, 65, 71, 73, 77, 80, 82, 85, 93]. Some researchers proposed to collect real-world LR-HR image pairs by using long-short camera focal lengths [4, 5, 69, 70, 74, 75, 88]; however, this is very costly and the trained models may only work well when similar photographing devices are used. Therefore, researchers propose to synthesize more realistic training data by designing more complex degradation models. The notable works include BSRGAN [84] and Real-ESRGAN [65]. In BSRGAN [84], Zhang et al. randomly shuffled and combined blur, downsampling and noise degradations to form a complex degradation, while in Real-ESRGAN [65], Wang et al. developed a high-order degradation model with several repeated degradation operations. Recently, researchers have also proposed to introduce human guidance into the training data generation process [6].

Given the training data with more realistic degradations, another issue is how to train the network to achieve the goal of Real-ISR. It is well-known that the $L_1$ or $L_2$ loss, which aims to minimize the fidelity error, often results in over-smoothed image details. To tackle this issue, in the past a few years, the generative adversarial networks (GANs) [18] have been widely adopted to train Real-ISR models [33, 35, 41, 46, 52, 66, 67, 87]. With the help of adversarial loss, GAN models can learn to find an image reconstruction path to generate more sharp details. Though great progress has been made, one critical limitation of GAN based Real-ISR models remain, *i.e.,* they incline to hallucinate visually unpleasant artifacts. Very recently, with the rapid development of diffusion models (DMs) [25, 59], it becomes popular to leverage the pre-trained large scale text-to-image models, such as stable diffusion (SD) [55], to achieve Real-ISR. Benefiting from the strong generative priors in DMs, some

recent works [56, 63, 79] have demonstrated encouraging Real-ISR results with fine-scale and realistic details. However, DMs have high randomness, which lead to unstable Real-ISR outputs and false image details.

In this paper, we aim to improve the GAN and DM based Real-ISR methods, reducing the artifacts and producing more realistic details, by proposing a new training loss function. It is well-known that natural images exhibit repetitive patterns across the whole image. Such a property of self-similarity has been extensively used in many image restoration algorithms, such as BM3D [11], NCSR [15], WNNM [21], and NLSN [49], where the image self-similarity is used as a prior to regularize the restored image. In this work, we employ the image self-similarity property as a powerful penalty to supervise the Real-ISR training progress. The proposed image self-similarity loss (SSL) could act as a plug-and-play penalty in most of the existing generative Real-ISR models, guiding them to exploit more effectively the inherent image self-similarity information for detail reconstruction. Specifically, we compute a self-similarity graph (SSG) to describe the image structural dependency, and minimize the distance between the SSGs of the ground-truth (GT) and Real-ISR output to optimize the model. To make the training process more efficient and focus more image edge/texture areas, we generate an edge mask from the GT image in an offline manner, and only build the SSG upon the edge pixels.

Our proposed SSL can be easily adopted into the off-the-shelf GAN-based and DM-based Real-ISR models as an extra penalty to enhance image details and reduce the unpleasant artifacts. An example is shown in Fig. 1. One can see that SwinIRGAN [40] over-smooths the image textures and generates wrong details, while the recent DM-based StableSR [63] restores much clearer details but still fails to generate some fine scale structures or correct textures. In comparison, the StableSR model trained with our SSL could reconstruct both clear content and more realistic textures with better perception quality. Our extensive experiments on state-of-the-art Real-ISR models validate the effectiveness of our proposed SSL, either in GAN-based or DM-based ISR tasks.

## 2 Related Work

**Traditional Fidelity-Oriented ISR.** Since SRCNN [13], many CNN backbones have been developed to promote the ISR performance in terms of PSNR and SSIM [68] measures. EDSR [42] and RDN [91] incorporate residual and densely connections, respectively. RCAN [90], OA-DNN [16], CRAN [92] and HAN [51] make use of channel/spatial attention modules. Recently, transformer-based models have shown stronger ability towards long-range dependency modeling. SwinIR [40] utilizes shifted partition windows to compute the image self-attention. ELAN [89] introduces multi-scale self-attention blocks to extract long-distance dependency. ACT [78] utilizes CNN to extract local interaction and transformer to obtain long-range dependency.

**GAN-based Generative Real-ISR.** The fidelity-oriented ISR models often generate over-smoothed details, sacrificing the perceptual quality of natural images. Inspired by GAN[18], many generative ISR methods have been proposed to obtain more photo-realistic results. SRGAN [33], and ESRGAN [67] utilize VGG-style [58] discriminator to perform the adversarial training. BSRGAN [84] and

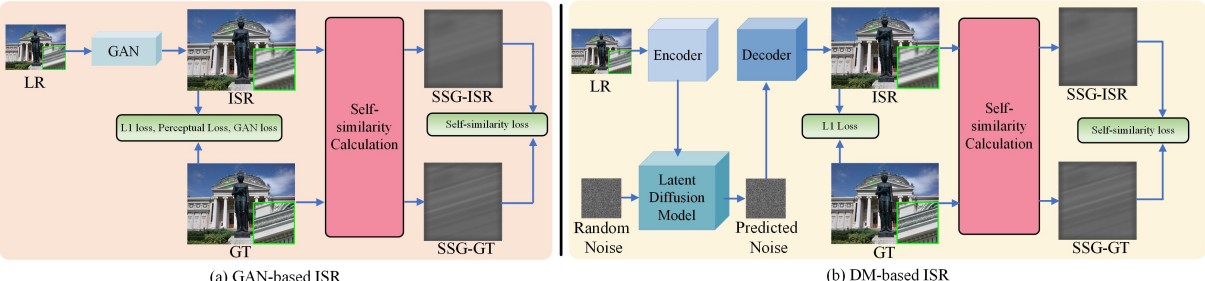

**Figure 2: Illustration of the training progress of (a) generative adversarial network (GAN) based and (b) latent diffusion model (DM) based Real-ISR by using our proposed self-similarity loss (SSL). The GAN or DM network is employed to map the input LR image to an ISR output. We calculate the self-similarity graphs (SSG) of both ISR output and ground-truth (GT) image, and calculate the SSL between them to supervise the generation of image details and structures.**

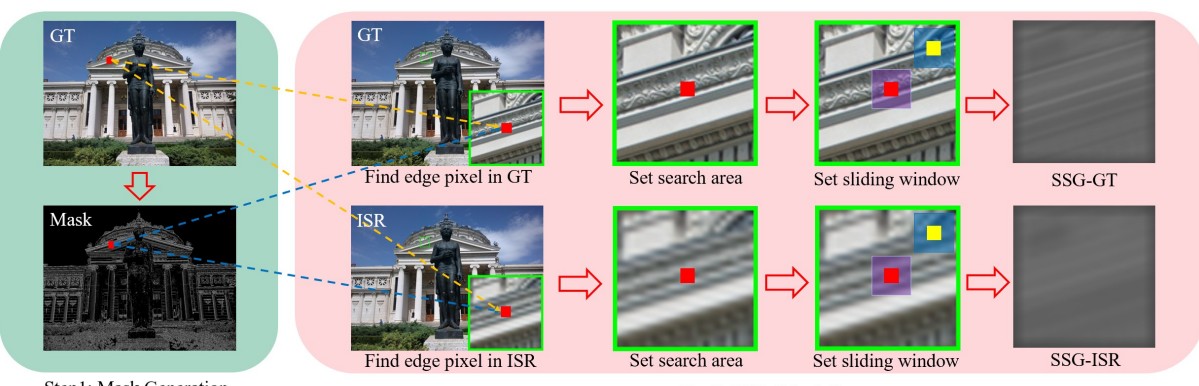

**Figure 3: Illustration of the self-similarity graph (SSG) computing process. We first generate a mask to indicate the image edge areas by applying the Laplacian Operator on the GT image. During the training period, for each edge pixel in the mask, we find the corresponding pixels in the GT image and ISR image, and set a search area centred at them. A local sliding window is utilized to calculate the similarity between each pixel in the search area and the central pixel so that an SSG can be respectively computed for the GT image and the ISR image, with which the SSL can be computed. The red pixel means the edge pixel, while the blue block means the sliding window.**

Real-ESRGAN [65] introduce complex degradation processes to synthesize the real-world degradation. HGGT [6] annotates positive and negative training pairs to enhance the perceptual quality in GAN training progress. In order to make the GAN training process more stable, SROBB [54] presents an enhanced perceptual loss to restrain the model with different semantic labels. SPSR [46] embeds the well-extracted structure prior into the RRDB network [67]. RankSRGAN [87] firstly trains a ranker model to indicate the relative perception quality of an image, and then utilizes a well-trained ranker to guide the generator to reconstruct better details. BebyGAN [35] searches the best candidate GT patch in the neighborhood to perform LR-HR supervision. LDL [41] computes an artifact map to indicate the local artifacts in ISR outputs, then imposes appropriate penalty on the artifact areas to improve the perceptual quality.

**DM-based Generative Real-ISR.** The powerful generative priors embedded in DMs can be exploited for Real-ISR. SR3 [56] utilizes a conditional pixel-level DM to iteratively denoise and generate

super-resolved results. StableSR [62] performs Real-ISR in the latent diffusion space by using a controllable feature wrapping module to balance between the reconstruction fidelity and the perceptual quality. PASD [76] introduces a pixel-aware cross attention block as a controllable module to guide the high-quality details generation. DiffBIR [43] utilizes a two-stage model, which first reduces complicated degradation factors and then uses the well-trained generative prior in SD to reconstruct delicate contents. DiffIR [72] pre-trains a DM with high-quality GT images to obtain abundant priors and then finetunes the DM with their low-quality counterparts to complete the Real-ISR task. ResShift [79] adopts an efficient sampling strategy through shifting the residual between high-quality and low-quality images to largely accelerate the diffusion steps.

## 3 Image Self-similarity Loss

The proposed training framework is illustrated in Fig. 2. In addition to the commonly used $L_1$, perceptual loss, adversarial loss in GAN-based methods, or the Gaussian noise prediction MSE loss in

**Table 1: Quantitative results of seven representative GAN-based Real-ISR models and their counterparts coupled with the proposed SSL. The bicubic degradation model is used here. For each of the seven groups of comparisons, the better results are highlighted in boldface. The PSNR and SSIM indices are computed in the Y channel of Ycbcr space.**

| Method | | ESRGAN | ESRGAN-SSL | RankSR GAN | RankSR GAN-SSL | SPSR | SPSR-SSL | Beby GAN | Beby GAN-SSL | LDL | LDL-SSL | ELAN GAN | ELAN GAN-SSL | SwinIR GAN | SwinIR GAN-SSL |
|---|---|---|---|---|---|---|---|---|---|---|---|---|---|---|---|
| Training Dataset | | DF2K_OST | | DIV2K | | DIV2K | | DF2K | | DF2K | | DIV2K | | DF2K | |
| Set5 | PSNR↑ | 30.4378 | **30.7786** | 29.6518 | **30.1834** | 30.3967 | **30.4476** | 30.4951 | **31.0239** | 31.0332 | **31.1106** | 30.7134 | **30.7942** | 31.0982 | **31.1616** |
| | SSIM↑ | 0.8523 | **0.8582** | 0.8379 | **0.8517** | 0.8443 | **0.8458** | 0.8550 | **0.8652** | 0.8611 | **0.8660** | 0.8526 | **0.8588** | 0.8686 | **0.8702** |
| | LPIPS↓ | 0.0739 | **0.0621** | 0.0699 | **0.0675** | 0.0615 | **0.0598** | 0.0595 | **0.0583** | 0.0660 | **0.0617** | 0.0547 | **0.0528** | 0.0682 | **0.0594** |
| | DISTS↓ | 0.0970 | **0.0929** | 0.1038 | **0.1014** | 0.0924 | **0.0906** | 0.0912 | **0.0906** | 0.0934 | **0.0924** | 0.0854 | **0.0840** | 0.0977 | **0.0936** |
| Set14 | PSNR↑ | 26.2786 | **26.7148** | 26.4514 | **26.6163** | **26.6423** | 26.6410 | 26.8625 | **27.1674** | 26.9378 | **27.1195** | 26.9128 | **27.0785** | 27.0486 | **27.3178** |
| | SSIM↑ | 0.6992 | **0.7114** | 0.7030 | **0.7132** | **0.7138** | 0.7091 | 0.7270 | **0.7272** | 0.7212 | **0.7263** | 0.7242 | **0.7287** | 0.7314 | **0.7385** |
| | LPIPS↓ | 0.1314 | **0.1202** | 0.1350 | **0.1337** | **0.1303** | 0.1331 | 0.1204 | **0.1200** | 0.1198 | **0.1169** | 0.1144 | **0.1123** | 0.1201 | **0.1114** |
| | DISTS↓ | 0.0985 | **0.0937** | 0.1104 | **0.1065** | 0.0990 | **0.0960** | **0.0930** | 0.0937 | **0.0917** | 0.0924 | 0.0940 | **0.0934** | 0.0983 | **0.0962** |
| DIV2K100 | PSNR↑ | 28.1983 | **28.7341** | 28.0314 | **28.3523** | 28.2042 | **28.5881** | 28.6301 | **29.1332** | 28.8401 | **29.0378** | 28.6631 | **28.8978** | 28.9873 | **29.3940** |
| | SSIM↑ | 0.7773 | **0.7896** | 0.7667 | **0.7822** | 0.7734 | **0.7849** | 0.7907 | **0.8012** | 0.7910 | **0.7985** | 0.7882 | **0.7959** | 0.8026 | **0.8118** |
| | LPIPS↓ | 0.1150 | **0.0995** | 0.1207 | **0.1143** | 0.1085 | **0.1021** | 0.1021 | **0.0974** | 0.0993 | **0.0952** | 0.0978 | **0.0936** | 0.0944 | **0.0911** |
| | DISTS↓ | 0.0594 | **0.0518** | 0.0637 | **0.0610** | 0.0541 | **0.0510** | **0.0491** | 0.0522 | 0.0522 | **0.0519** | **0.0494** | 0.0499 | 0.0496 | **0.0490** |
| Urban100 | PSNR↑ | 24.3548 | **25.2991** | 24.4686 | **24.5959** | 24.7978 | **25.2672** | 25.2205 | **25.6060** | 25.4537 | **25.5851** | 25.5041 | **25.7764** | 25.8311 | **26.2520** |
| | SSIM↑ | 0.7340 | **0.7606** | 0.7294 | **0.7366** | 0.7473 | **0.7573** | 0.7628 | **0.7696** | 0.7661 | **0.7705** | 0.7696 | **0.7761** | 0.7850 | **0.7929** |
| | LPIPS↓ | 0.1234 | **0.1061** | 0.1381 | **0.1287** | 0.1186 | **0.1087** | 0.1094 | **0.1048** | 0.1084 | **0.1037** | 0.1053 | **0.1007** | 0.0998 | **0.0941** |
| | DISTS↓ | 0.0879 | **0.0811** | 0.1044 | **0.1010** | 0.0850 | **0.0809** | 0.0797 | **0.0786** | 0.0793 | **0.0785** | 0.0805 | **0.0788** | 0.0807 | **0.0783** |
| BSDS100 | PSNR↑ | 25.3277 | **25.7504** | 25.4646 | **25.5562** | 25.5092 | **25.6818** | 25.7918 | **26.1428** | 25.9741 | **26.0676** | 25.7897 | **25.9174** | 26.1063 | **26.2676** |
| | SSIM↑ | 0.6533 | **0.6723** | 0.6510 | **0.6598** | 0.6599 | **0.6649** | 0.6791 | **0.6849** | 0.6818 | **0.6870** | 0.6731 | **0.6768** | 0.6908 | **0.6957** |
| | LPIPS↓ | 0.1601 | **0.1477** | 0.1736 | **0.1675** | 0.1602 | **0.1568** | 0.1505 | **0.1490** | 0.1534 | **0.1469** | 0.1492 | **0.1436** | 0.1574 | **0.1450** |
| | DISTS↓ | 0.1173 | **0.1148** | 0.1280 | **0.1274** | 0.1186 | **0.1160** | **0.1137** | 0.1155 | 0.1175 | **0.1160** | **0.1113** | 0.1122 | 0.1168 | **0.1126** |
| Manga109 | PSNR↑ | 28.4125 | **29.2324** | 27.8481 | **28.2400** | 28.5608 | **28.9309** | 29.1934 | **29.7105** | 29.6204 | **29.7949** | 29.2020 | **29.4077** | 29.8802 | **30.2567** |
| | SSIM↑ | 0.8595 | **0.8697** | 0.8497 | **0.8583** | 0.8591 | **0.8618** | 0.8754 | **0.8794** | 0.8734 | **0.8807** | 0.8698 | **0.8753** | 0.8892 | **0.8935** |
| | LPIPS↓ | 0.0644 | **0.0566** | 0.0754 | **0.0690** | 0.0663 | **0.0622** | 0.0524 | **0.0520** | 0.0540 | **0.0502** | 0.0577 | **0.0532** | 0.0469 | **0.0440** |
| | DISTS↓ | 0.0468 | **0.0403** | 0.0577 | **0.0564** | 0.0460 | **0.0454** | **0.0355** | 0.0378 | 0.0354 | **0.0353** | 0.0436 | **0.0429** | **0.0341** | 0.0345 |
| General100 | PSNR↑ | 29.4251 | **30.0092** | 29.1108 | **29.4338** | 29.4237 | **29.7783** | 29.9510 | **30.3979** | 30.2891 | **30.4594** | 29.9434 | **29.9606** | 30.4339 | **30.4839** |
| | SSIM↑ | 0.8095 | **0.8215** | 0.8017 | **0.8122** | 0.8091 | **0.8160** | 0.8222 | **0.8317** | 0.8280 | **0.8330** | 0.8220 | **0.8230** | 0.8352 | **0.8370** |
| | LPIPS↓ | 0.0878 | **0.0795** | 0.0954 | **0.0908** | 0.0863 | **0.0809** | 0.0780 | **0.0761** | 0.0800 | **0.0760** | 0.0768 | **0.0760** | 0.0765 | **0.0727** |
| | DISTS↓ | 0.0877 | **0.0832** | 0.0977 | **0.0948** | 0.0890 | **0.0859** | **0.0801** | 0.0814 | 0.0806 | **0.0796** | **0.0817** | 0.0826 | **0.0824** | 0.0826 |

DM-based methods, we calculate the self-similarity graphs (SSG) of both ISR output and ground-truth (GT), and consequently introduce a self-similarity loss (SSL) between them to supervise the reconstruction of image details and structures.

## 3.1 Image Self-similarity

For a natural image, one can observe many repetitive patterns across it, known as the image self-similarity. Such a property has been used to improve the image restoration performance for a long time [3, 11]. Actually, the self-attention mechanism [44, 61] in transformer models exploits the image self-similarity in deep feature space. In this paper, we adopt the Exponential Euclidean distance [3] to calculate the self-similarity. For any two patches $I_p, I_q \in \mathbb{R}^{(2f+1) \times (2f+1) \times C}$ centered at pixels $\mu_p$ and $\mu_q$ in image $I \in \mathbb{R}^{H \times W \times C}$, respectively, where $f$ denotes the patch radius, $H$, $W$ and $C$ are the image height, width and channel number ($C$=3 for RGB images), we firstly compute the squared Euclidean distance between $I_p$ and $I_q$:

$$d^2(I_p, I_q) = \frac{1}{C(2f+1)^2} \sum_{i=1}^{C} \sum_{j=-f}^{f} (\mu_{p+j}^i - \mu_{q+j}^i)^2, \quad (1)$$

where $\mu_{p+j}^i$ and $\mu_{q+j}^i$ denote the neighborhood pixels around $\mu_p^i$ and $\mu_q^i$ in patch $I_p$ and $I_q$, respectively. The similarity $S(I_p, I_q)$ between $I_p$ and $I_q$ is calculated as:

$$S(I_p, I_q) = e^{-\frac{d^2(I_p, I_q)}{h}}, \quad (2)$$

where $h > 0$ is a scaling factor. One can see that $0 \le S(I_p, I_q) \le 1$. When the Euclidean distance $d^2(I_p, I_q)$ approaches to 0, the similarity $S(I_p, I_q)$ approaches to 1, indicating that the two patches are highly similar.

## 3.2 Mask Generation

By using the self-similarity measure defined in Eq. 2, we could compute the similarity of a patch with all the other patches in the whole image, and construct a self-similarity graph (SSG). However, this is computationally expensive because the size of such an SSG will be $H^2 \times W^2$. Actually, we do not need to calculate the self-similarity for each patch since the challenges of Real-ISR lie in edge and texture areas instead of smooth regions. Therefore, we can generate a mask of edge/texture pixels to indicate where we should calculate the SSG. For simplicity, we first generate an edge map $E \in \mathbb{R}^{H \times W}$ by applying the Laplacian operator, denoted by $L$, to the GT image $I_{HR} \in \mathbb{R}^{H \times W \times C}$, i.e., $E = L \otimes I_{HR}$. Then, we obtain the binary mask $M \in \mathbb{R}^{H \times W}$ by thresholding $E$:

$$M_{i,j} = \begin{cases} 0, & E_{i,j} \le t \\ 1, & E_{i,j} > t \end{cases} \quad (3)$$

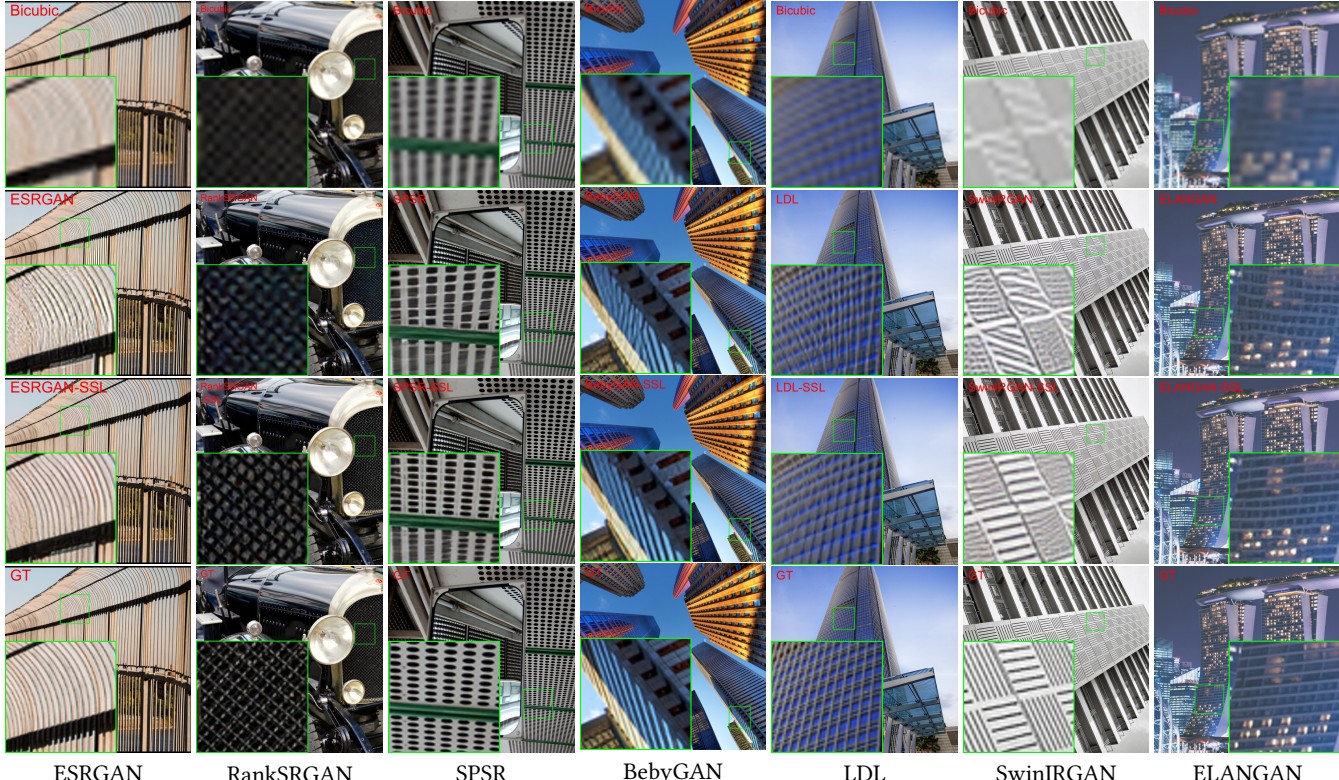

ESRGAN          RankSRGAN          SPSR          BebyGAN          LDL          SwinIRGAN          ELANGAN

**Figure 4: Visual comparison of the state-of-the-art GAN based Real-ISR models and their counterparts trained with our SSL. The bicubic degradation model is used here. From the top row to the bottom row are the results of bicubic interpolation, the original Real-ISR model, the Real-ISR model trained with our SSL, and the GT image. Please zoom in for better observation.**

where $t$ is a threshold. We empirically set it to 20 to retain most of the true edge pixels while filtering out smooth and trivial image features. $M$ is computed in an off-line manner to avoid the repetitive computation in each iteration.

In the training progress, for pixels at $(i, j)$ where $M_{i,j} = 1$, we find the corresponding RGB pixels $\mu_p$ in the GT image and the ISR output, and calculate their SSG for comparison. On the DF2K_OST training dataset, the edge pixels occupy only 13% of the total amount of image pixels. By using $M$ to guide the construction of SSG, we can not only reduce significantly the training cost, but also concentrate on the image edges and textures.

### 3.3 Self-similarity Graph Calculation

For an edge pixel $p$ in the original RGB image $I$ (the corresponding pixel in the Mask $M$ is $M_p = 1$), we define a search area $I_{K_s} \in \mathbb{R}^{K_s \times K_s \times C}$ as well as a local window $I_p \in \mathbb{R}^{K_w \times K_w \times C}$ centered at it, where $K_w = 2f + 1$ and $f$ is the radius of the window. Then for each pixel $q$ in the search area, we extract a sliding window $I_q \in \mathbb{R}^{K_w \times K_w \times C}$ to calculate its similarity with $I_p$, i.e., $S(I_p, I_q)$, by Eq. 1 and Eq. 2. Then we normalize $S(I_p, I_q)$ as:

$$\bar{S}(I_p, I_q) = \frac{1}{\epsilon} * S(I_p, I_q), \tag{4}$$

where $\epsilon = \sum_{q \in I_{K_s}} S(I_p, I_q)$ is the normalization factor.

The overall calculation procedure of SSG is illustrated in Fig. 3. To be more specific, for each edge pixel in the mask, we find the corresponding pixels in the GT image and ISR image, and set a search area centred at them. Then we set a local sliding window to calculate the similarity between the patch centered at the central pixel and another patch centered at the pixels in the search area. All the values of $\bar{S}(I_p, I_q)$ builds the SSG of image $I$, which describes the inherent structural similarity distribution of the image. In practice, we could sample $I_q$ with a stride $s$ to further reduce the computational cost (we set $s = 3$ in our implementation).

### 3.4 Self-similarity Loss

Denote by $\bar{S}_{HR}$ and $\bar{S}_{ISR}$ the SSG of the GT image and the ISR output, respectively. We can use their distance as the loss to supervise the network training. Here we employ the KL-divergence and $L_1$ distance to build the SSL:

$$L_{SSL} = D_{KL}(\bar{S}_{HR} || \bar{S}_{SR}) + \alpha |\bar{S}_{SR} - \bar{S}_{HR}|, \tag{5}$$

where $\alpha$ is a balance parameter and we simply set it as 1 in all our experiments.

**SSL in GAN-based Models.** To apply SSL into an off-the-shelf GAN-based Real-ISR method, we just need to add the above $L_{SSL}$ loss to its original loss function $L_{original}$ (e.g. pixel-wise $L1$ loss,

**Table 2: Quantitative results of three representative DM-based Real-ISR models and their counterparts coupled with the proposed SSL. For each of the three groups of comparisons, the better results are highlighted in boldface. The PSNR and SSIM indices are computed in the Y channel of Ycbcr space.**

| Method | | StableSR | StableSR-SSL | ResShift | ResShif-SSL | DiffIR | DiffIR-SSL |
|---|---|---|---|---|---|---|---|
| Training Dataset | | DF2K_OST+DIV8K+FFHQ | | | | | |
| DIV2K100 | PSNR↑ | **23.2988** | 23.1111 | 23.6136 | **24.7275** | **25.5008** | 25.4124 |
| | SSIM↑ | **0.5654** | 0.5203 | 0.5701 | **0.6161** | **0.6570** | 0.6470 |
| | LPIPS↓ | **0.3125** | 0.3588 | 0.3712 | **0.3417** | **0.2651** | 0.2664 |
| | DISTS↓ | **0.2045** | 0.2323 | 0.2379 | **0.2236** | 0.2013 | **0.1977** |
| | FID↓ | **24.4578** | 28.0564 | 49.3542 | **35.4661** | 25.7638 | 26.1045 |
| | NIQE↓ | 4.7806 | **4.5219** | **6.2656** | 6.6347 | 5.1936 | **4.9569** |
| | CLIP-IQA↑ | 0.6694 | **0.6940** | **0.6859** | 0.5343 | 0.5130 | **0.5262** |
| | MUSIQ↑ | 65.7710 | **67.7485** | **64.0147** | 57.1369 | 58.5725 | **60.8936** |
| DRealSR | PSNR↑ | **28.1526** | 27.6065 | 27.5799 | **29.4468** | **29.9046** | 29.5164 |
| | SSIM↑ | **0.7529** | 0.6736 | 0.7364 | **0.7975** | **0.8188** | 0.8171 |
| | LPIPS↓ | **0.3315** | 0.4312 | 0.3941 | **0.3818** | 0.2895 | **0.2683** |
| | DISTS↓ | **0.2263** | 0.2810 | 0.2709 | **0.2684** | 0.2118 | **0.2031** |
| | FID↓ | **151.1807** | 156.5795 | 180.2996 | **161.8707** | 141.2654 | 141.9467 |
| | NIQE↓ | 6.5808 | **6.2259** | **7.0837** | 9.0121 | **7.1668** | 7.5738 |
| | CLIP-IQA↑ | 0.6207 | **0.6103** | **0.6490** | 0.3922 | 0.3191 | **0.3722** |
| | MUSIQ↑ | 58.4207 | **60.5404** | **58.4495** | 38.3946 | 41.9419 | **45.0333** |
| RealSR | PSNR↑ | 24.7021 | **25.3236** | 25.5153 | **26.6132** | **27.4546** | 26.9851 |
| | SSIM↑ | **0.7065** | 0.6563 | 0.7024 | **0.7493** | **0.7848** | 0.7847 |
| | LPIPS↓ | **0.3018** | 0.3744 | 0.3759 | **0.3446** | 0.2538 | **0.2411** |
| | DISTS↓ | **0.2135** | 0.2496 | 0.2725 | **0.2552** | 0.1928 | **0.1892** |
| | FID↓ | **129.5313** | 131.4425 | 164.7055 | **149.3277** | 117.8279 | 119.3884 |
| | NIQE↓ | 5.9430 | **5.1812** | **6.3815** | 7.0816 | 6.4811 | **6.2603** |
| | CLIP-IQA↑ | 0.6178 | **0.6324** | **0.6838** | 0.4760 | 0.3437 | **0.3677** |
| | MUSIQ↑ | **65.7834** | 65.6814 | **63.2059** | 52.8008 | 52.0040 | **54.8398** |
| DPED-iPhone | NIQE↓ | 6.7597 | **6.1154** | **8.9634** | 9.4691 | 7.2465 | **7.1488** |
| | CLIP-IQA↑ | 0.4694 | **0.4944** | **0.6174** | 0.4109 | 0.2664 | **0.3045** |
| | MUSIQ↑ | 50.6582 | **53.7349** | **47.8474** | 37.6449 | 36.4193 | **41.2825** |

perceptual loss and GAN loss), and then re-train the model:

$$L_{total} = L_{original} + \beta L_{SSL}, \qquad (6)$$

where $\beta$ is a balance parameter.

**SSL in DM-based Models.** As for those latent DM-based Real-ISR methods, StableSR [63] and ResShift [79], the $L_{original}$ is applied to predict the desired noise in latent space. Since SSL is computed in image space, we need to pass the predicted noise through the VAE decoder to output the ISR image, as shown in Fig. 2(b), and then apply the SSL to the reconstructed image. We also employ a pixel-wise $L_1$ loss for more stable training. The total loss is:

$$L_{total} = L_{original} + \beta L_{SSL} + \gamma L_1, \qquad (7)$$

where $\beta$, $\gamma$ are balance parameters. The $L_{SSL}$ and $L_1$ will back-propagate their gradients to update the parameters of the denoising UNet and the controlling parts in DMs.

## 4 Experimental Results

### 4.1 Experiments on GAN-based Models

**Comparison Methods.** Our proposed SSL can be directly applied to the existing GAN-based Real-ISR models with either simple bicubic degradation or complex mixture degradations [65, 84] as a plug-and-play module to improve their performance. For bicubic degradation, we embed SSL into ESRGAN [67], RankSRGAN [87], SPSR [46], BebyGAN [35] and LDL [41]. For complex mixture degradations, we embed SSL into Real-ESRGAN [65] and BSRGAN [84]. Most of the above models employ the CNN backbone (*e.g.*, RRDB [67] or SRResNet [33]) as the generator. In this paper, we also employ the transformer backbones, *i.e.*, SwinIR [40] and ELAN [89], as the generator, resulting in the SwinIRGAN and ELANGAN models. For each of the above Real-ISR models (*e.g.*, ESRGAN), we denote by "*-SSL" (*e.g.*, ESRGAN-SSL).

**Training Details**. For each of the evaluated Real-ISR methods, we train its SSL guided counterpart with the same patch size and training dataset (*i.e.*, DIV2K [60], DF2K [1, 60] and DF2K-OST [1, 60, 66]) as the original method. In the experiments with complex degradations, since the original degradation setting in Real-ESRGAN and BSRGAN is too heavy, we follow the Real-ESRGAN and BSRGAN settings in HGGT [6] (which has weaker degradation level) for training data generation. The Adam [32] optimizer is adopted. The initial learning rate is set to 1e-4, which is halved after 200K iterations for CNN backbones, and 200K, 250K, 275K, 287.5K

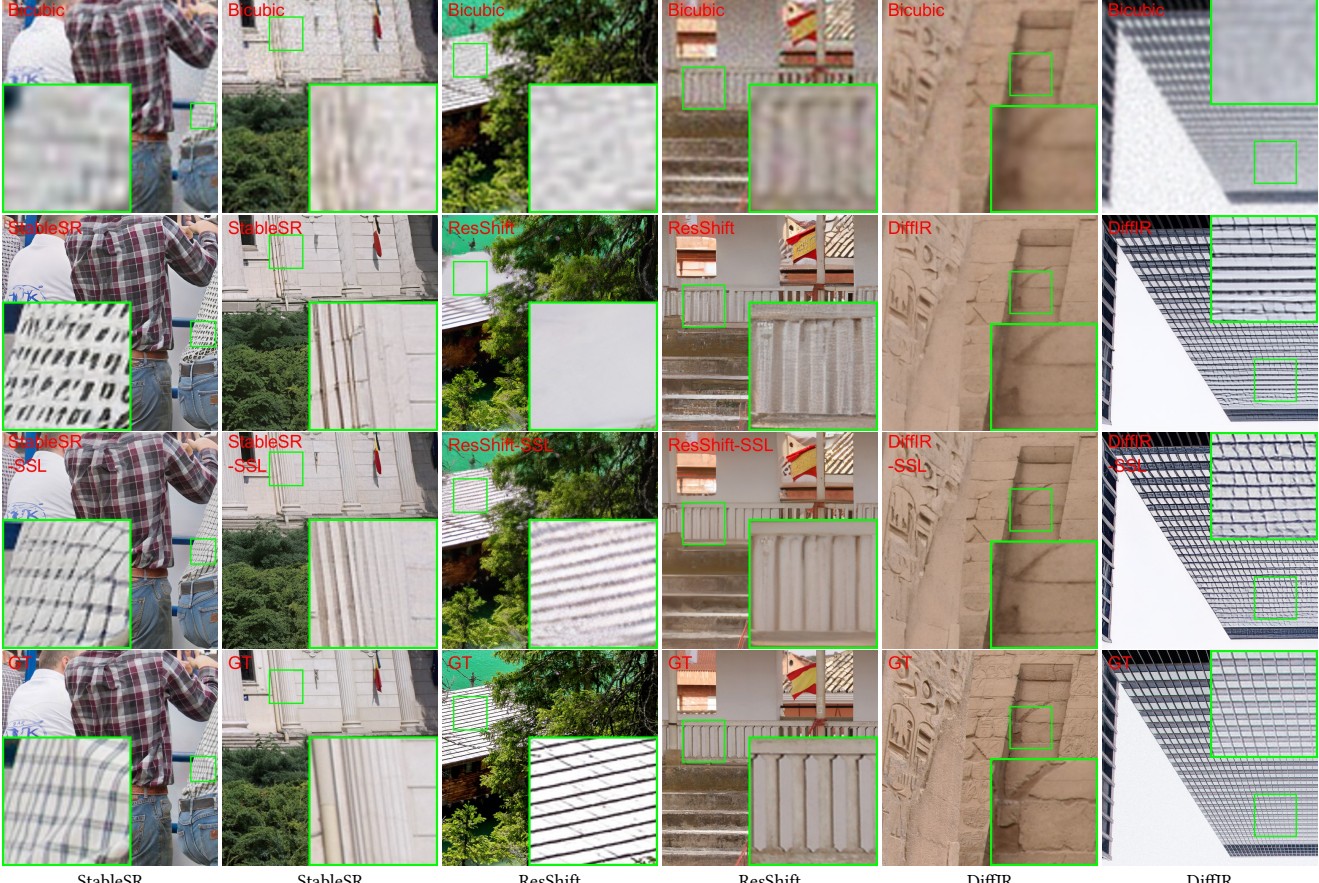

**Figure 5: Visual comparison of the state-of-the-art DM based Real-ISR models and their counterparts trained with our SSL. From the top row to the bottom row are the results of bicubic interpolation, the original Real-ISR model, the Real-ISR model trained with our SSL, and the GT image. Please zoom in for better observation.**

iterations for transformer backbones. When calculating SSG, the search area $I_{K_s}$ is set to 25, the sliding window $I_{K_w}$ is set to 9, and the scaling factor $h$ is set to 0.004. $\beta$ is set to 1000. All experiments are conducted on NVIDIA RTX 3090 GPUs. All of the SSL guided models are fine-tuned from a well-trained fidelity-oriented version (e.g. RRDB [67], SwinIR [40] or ELAN [89]which is trained only with $L1$ loss without a discriminator) for better initialization.

**Evaluation Datasets and Metrics**. We employ the widely-used testing benchmarks, including Set5 [2], Set14 [81], DIV2K100 [60], Urban100 [26], BSDS100 [47], Manga109 [48], General100 [14], to evaluate the competing methods. Considering the fact that there is certain randomness in the synthesis of LR images when using the complex mixture degradation models in [65, 84], for each test image, we synthesize a group of 30 LR images using randomly sampled degradation factors, and report the averaged metrics for fair and solid evaluation. We compute PSNR and SSIM [68] in the Y channel for fidelity measurement. For perceptual quality, LPIPS [86] and DISTS [12] are used for quantitative assessment.

**Results for Bicubic Degradation**. Table 1 shows the quantitative results of different Real-ISR models when bicubic degradation is used. It can be seen that on all the 7 testing datasets, our SSL

guided models surpass their original counterparts in most of the fidelity (PSNR, SSIM) and perceptual (LPIPS, DISTS) measures, no matter the CNN or transformer backbones are used. This demonstrates that the image SSG could characterize the image inherent structures, and our SSL could provide effective supervision in the Real-ISR model training process, enforcing the models to hallucinate more correct contents with better fidelity and suppressing the visual artifacts to achieve better perceptual quality. It is worth mentioning that our SSL will not introduce any extra cost in the inference process.

**Qualitative Result.** Fig .4 provides visual comparisons between major Real-ISR models and their SSL-guided version in the case of bicubic degradation. One can clearly see that the SSL-guided models could generate clearer textures (the 1st column), or richer details (the 2nd column) and correct the twisted textures (the 3rd/4th/5th/6th/7th columns) generated by the original models. Such observations echo with the results in Table 1, proving again that SSL could hallucinate correct details and suppress artifacts.

**Results for Complex Degradation.** Due to the page limit, we provide the quantitative results of GAN-based Real-ISR models and their SSL guided versions under complex image degradation, as

well as the visualization comparisons between them, in the **supplementary material**.

## 4.2 Experiments on DM-based Models

**Comparison Methods**. We embed SSL into three representative DM-based models, including StableSR [62], ResShift [79] and DiffIR [72]. For each of the above Real-ISR models (*e.g.*, StableSR), we denote by "*-SSL" (*StableSR-SSL*).

**Training Details.** For each of the evaluated DM-based Real-ISR method, we employ the same training datasets (including DF2K-OST [1, 60, 66], DIV8K [20], FFHQ [29]), and apply the same degradation pipeline as that used in StableSR [62]. The training patch size and iterations in SSL-guided versions are set to the same as the original method. The Adam [32] optimizer is used. The learning rate is fixed to 5e-5. When calculating SSG, the search area $I_{K_s}$ is set to 25, the sliding window $I_{K_w}$ is set to 9, and the scaling factor $h$ is set to 0.004. For SSL guided StableSR and DiffIR, the weight $\beta$ and $\gamma$ in Eq. 7 are set to 1 and 0.1, respectively. For SSL guided DiffIR, since the original model [72] already utilizes pixel-wise $L1$ loss, then we implement the loss function type as Eq. 6, $\beta$ is set to 1000. All experiments are conducted on NVIDIA V100 GPUs. We update all of the parameters of UNet in the pre-trained DM as well as the controlling parts towards the SSL-guided counterparts.

**Evaluation Datasets and Metrics**. We utilize the testing images from StableSR [62], including the 3000 synthesized DIV2K100 low-quality testing images (each GT image has a group of 30 LR images generated from DIV2K100 [60] dataset with complicated degradation factors), RealSR [4] (100 real-world low-quality images with their corresponding GTs obtained by cameras), DRealSR [70] (93 real-world low-quality images with their corresponding GTs captured by cameras), DPED-iphone [28] (113 real-world low-quality images taken by iPhone without GT). We compute full-reference image quality metrics, including PSNR, SSIM [68], LPIPS [86] and DISTS [12], and no-reference image quality metrics, including NIQE [50], CLIP-IQA [62] and MUSIQ [30]. The statistical distance metric FID [24] is also calculated.

**Quantitative Result.** Tab. 2 shows the numerical results of the original DM-based Real-ISR methods and their SSL guided versions. One can see that StableSR-SSL obtains better no-reference metrics (NIQE/CLIP-IQA/MUSIQ) while gets worse full-reference metrics (PSNR/SSIM/LPIPS/DISTS). ResShift-SSL gets better full-reference metrics (PSNR/SSIM/LPIPS/DISTS) but worse no-reference metrics (NIQE/CLIP-IQA/MUSIQ). DiffIR-SSL gets better perceptual-relevant metrics (LPIPS/DISTS/NIQE/CLIP-IQA/MUSIQ). While different SSL guided models obtain different performance, it is still reasonable for the following reasons: (1). StableSR-SSL leverages a pretrained Stable-Diffusion model [55], which is trained on LAION-5B [57], a multi-modal dataset that contains a vast number of text-to-image pairs. This results in a distinct data distribution divergence when compared to the general training datasets used for SR tasks, such as DF2K [1, 60] and DIV8K [20]. Consequently, during inference stage, the results generated by StableSR-SSL, exhibit a notable difference from the GT in the test set (such as DIV2K100). Thus all full-reference metrics get failure, while get much better no-reference metrics, which also indicate better perception quality. (2) ResShift-SSL gets better FR-IQA (PSNR/SSIM/LPIPS/DISTS/FID)

results, this indicated that SSL could help ResShift reconstruct textures with higher fidelity. As for the worse NR-IQA metrics, this is mainly because the existing NR-IQA metrics, including NIQE, CLIP-IQA and MUSIQ, favor the images with more high-frequency details, even these details are wrong. As can be seen from Fig. 6 in the **supplementary**, ResShift hallucinates many wrong details (*e.g.*, on the windows in column 1), while ResShift-SSL successfully removes those artifacts. Our user study in Fig. **??** also showed that 73.07% of the observers pick the results of ResShift-SSL. However, the NR-IQA metrics prefer the results of ResShift because they are not accurate enough to represent the human perception yet. (3). DiffIR-SSL not only trains a latent-diffusion model from scratch, but also utilizes a discriminator. Due to the influence introduced by the discriminator, the PSNR/SSIM just get worse, but obtains better perception-relevant metrics (LPIPS/DISTS/NIQE/CLIP-IQA/MUSIQ).

**Qualitative Result.** Fig. 5 shows the visualization results. One can see that compared with the original DM-based Real-ISR methods, their SSL guided versions perform significantly better in restoring the image structures and details, demonstrating the strong structure regularization capability of SSL. For example, StableSR generates false patterns in the T-shirt (column 1) and incomplete details on the peristyle (column 2), while the SSL guided StableSR restores correct T-shirt pattern and hallucinates more complete structures on peristyle. For ResShift, it either over-smooths the details (column 3) or generate wrong textures (column 4), while SSL can help to solve this issue. Similar observations go to DiffIR. All these results validate the effectiveness of SSL in encouraging the Real-ISR model to generate more delicate details. More visual comparisons that validate the perception imporvement of SSL can be found in the **supplementary material**.

Due to the limited pages, we also provide the following contents in our **supplementary material**: (1). A user study to validate the performance of SSL. (2). The training cost analysis towards SSL. (3). Ablation studies about the selection of hyper-parameters, such as $K_s$, $K_w$, $\beta$ in Eq. 6 and Eq. 7. (4). Limitation of SSL.

## 5 Conclusion

Generative image super-resolution methods, including GAN-based and DM-based ones, are prone to generating visual artifacts. In this work, we proposed a novel use of the image self-similarity prior for improving the generative real-world image super-resolution results. Specifically, we explicitly computed the self-similarity graph (SSG) of the image, and took the difference between the SSG maps of ground-truth image and Real-ISR output as a self-similarity loss (SSL) to supervise the network training. SSL could be easily embedded in off-the-shelf Real-ISR models, including GAN-based and DM-based ones, as a plug-and-play penalty, guiding the model to more stably generate realistic details and suppress false generations and visual artifacts. Our extensive experiments on benchmark datasets validated the generality and effectiveness of the proposed SSL in generative Real-ISR tasks.

## Acknowledgments

We faithfully thank PolyU-OPPO Joint Innovation Lab for supporting our projects.

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
