# OpenReview forum: "SSL: A Self-similarity Loss for Improving Generative Image Super-resolution"
_acmmm.org/ACMMM/2024/Conference — MM2024 Poster_

### Official Review · Reviewer_BV8i · 2024-05-22

**Rating:** 4
**Confidence:** 3

**Summary:**

This paper proposes a self-similarity loss (SSL) to improve the performance of generative models for real-world image super-resolution (Real-ISR), such as GANs and DMs. Natural images exhibit self-similarity property - local patches can have similar patches in the whole image. The paper computes self-similarity graphs (SSG) to describe this property for ground-truth and Real-ISR images. An edge mask is used to calculate SSG on edge areas only. SSL minimizes the distance between SSGs to supervise detail/structure generation and reduce artifacts. Experiments show the effectiveness of SSL in enhancing state-of-the-art generative Real-ISR models.

**Strengths:**

-	The self-similarity property is well-motivated and has been successfully used in image restoration before. Leveraging it for Real-ISR training is reasonable.
-	The edge mask and SSG computation make the method efficient and focus on important areas.
-	The SSL can be easily added to existing Real-ISR models as a plug-and-play module.
-	Comprehensive experiments are conducted on benchmark datasets to validate the method.

**Limitations:**

- Lack of comparison with some SOTA methods, like FeMaSR and DeSRA.
- How about adopting other edge detection operations rather than Laplacian operator.
- Lack of potential limitations of SSL

**Suitability:**

2

---

### Official Review · Reviewer_HewD · 2024-05-23

**Rating:** 5
**Confidence:** 3

**Summary:**

This paper aims at improving image super-resolution with the self-similarity loss. The self-similarity is one characteristic of images. The proposed self-similarity loss can be plugged into existing GAN-based or Diffusion-based Image Resolution solutions. Extensive experiments are performed to demonstrate the effectiveness of the proposed method.

**Strengths:**

1. Identify the artifacts of current image super-resolution techniques. Current methods generate either false structures or over-smoothed results. The proposed method generates more faithful details while suppressing the artifacts.
2. Explore the choice of several parameters of self-similarity calculation including the K_w and K_s which denotes the radius of the window and the search area. Also perform experiments to explore the new loss weight in GAN and Diffusion driven techniques.
3. In order to avoid naively applying the self-similarity on all pixels for one image, authors propose to exploit the mask at the edge area to selectively calculate important self-similarity of pixels.

**Limitations:**

1. No further analysis of mask choice. Authors use edge masks to reduce the computation cost. Are all pixels at the edges the best to calculate self-similarity? In other words, how do we define the good pixels of self-similarity loss to improve the image super-resolution.
2. Lack the experiments to verify whether the proposed self-similarity loss can benefit unsupervised image super-resolution solutions.
3. The proposed method analyzes several parameters like radius of the window, search area, loss weight setting. Are these parameters related to the image resolution or other loss levels? In the multi-loss items, what is the characteristic of the best parameter setting? With this kind of analysis, we can further understand how to set the tunable parameters.

**Suitability:**

2

---

### Official Review · Reviewer_eWxV · 2024-05-24

**Rating:** 4
**Confidence:** 3

**Summary:**

The authors propose a novel loss function which can improve the performance of image super resolution models. It take advantage of the self similarity within images to produce more accurate textures (typically repeated patterns and around edges). They compute this self-similarity loss by computing a scene graph on image patches and comparing the distribution. This loss produces more realistic textures in the reconstructions and improves quantitative image comparison metrics. Performance is also validated by a user study.

**Strengths:**

* Creative and novel solution
* Extensive evaluation
* Good results

**Limitations:**

* The explanations for worse quantitative performance on the diffusion-based methods is weak. I find these claims are made without any evidence to back it up and the explanations are not sufficient, for instance why performance on ResShift-SSL has consistently worse performance in NQIE, CLIP-IQA, and MUSIQ
* No training time comparison with/without their metric. As computing the proposed metric requires many calculations the affect on training time should be examined.
* I would appreciate to have more comparisons on smoother textures, to ensure training with the proposed does not reduce performance on those areas.

**Suitability:**

3

---

### Meta-Review · Area_Chair_Qrdh · 2024-06-30

**Recommendation:** Accept (Poster)
**Confidence:** 4

**Metareview:**

The final ratings are 3 weak accept. The reviewers acknowledged sufficient novelty and extensive evaluation. Initial concerns seemed to be resolved mostly through rebuttal.